# Integrating multiple plant functional traits to predict ecosystem productivity

Pu Yan[1,2,3], Nianpeng He ![ORCID] [1,2,4✉], Kailiang Yu[5,6], Li Xu[1,2,7] & Koenraad Van Meerbeek[3,8]

Quantifying and predicting variation in gross primary productivity (GPP) is important for accurate assessment of the ecosystem carbon budget under global change. Scaling traits to community scales for predicting ecosystem functions (i.e., GPP) remain challenging, while it is promising and well appreciated with the rapid development of trait-based ecology. In this study, we aim to integrate multiple plant traits with the recently developed trait-based productivity (TBP) theory, verify it via Bayesian structural equation modeling (SEM) and complementary independent effect analysis. We further distinguish the relative importance of different traits in explaining the variation in GPP. We apply the TBP theory based on plant community traits to a multi-trait dataset containing more than 13,000 measurements of approximately 2,500 species in Chinese forest and grassland systems. Remarkably, our SEM accurately predicts variation in annual and monthly GPP across China ($R^2$ values of 0.87 and 0.73, respectively). Plant community traits play a key role. This study shows that integrating multiple plant functional traits into the TBP theory strengthens the quantification of ecosystem primary productivity variability and further advances understanding of the trait-productivity relationship. Our findings facilitate integration of the growing plant trait data into future ecological models.

[1] Key Laboratory of Ecosystem Network Observation and Modeling, Institute of Geographic Sciences and Natural Resources Research, Chinese Academy of Sciences, Beijing 100101, China. [2] College of Resources and Environment, University of Chinese Academy of Sciences, Beijing 100049, China. [3] Division Forest, Nature and Landscape, Department of Earth and Environmental Sciences, KU Leuven, Leuven, Belgium. [4] Center for Ecological Research, Northeast Forestry University, Harbin 150040, China. [5] Department of Ecology and Evolutionary Biology, Princeton University, Princeton, NJ, USA. [6] High Meadows Environmental Institute, Princeton University, Princeton, NJ, USA. [7] Earth Critical Zone and Flux Research Station of Xing'an Mountains, Chinese Academy of Sciences, Daxing'anling 165200, China. [8] KU Leuven Plant Institute, KU Leuven, Leuven, Belgium. ✉email: henp@igsnrr.ac.cn

Gross primary productivity (GPP) is the largest carbon flux in terrestrial ecosystems and thus plays a prominent role in global carbon cycle regulation[1]. However, accurate prediction of GPP across ecosystems or regions remains challenging[2], especially in the context of climate change and the impact of human activities on the carbon balance of the biosphere[3]. Although much research has been conducted on how environmental changes across space and time affect ecosystem primary productivity[1,4,5], there remain fundamental knowledge gaps in terms of accurately capturing spatial or temporal variation in GPP and assessing its drivers. Intuitively, plants, especially its leaves, are the direct organs for photosynthesis and plant biomass production from first principles; hence, as the primary producer, plants contribute most to carbon fluxes across ecosystems[1]. Thus, plant leaf traits that are closely related to photosynthesis should directly influence ecosystem GPP, in combination with the direct and indirect effects of environmental factors, thereby regulating the global terrestrial carbon cycle and its response to climate change[6–8]. In this sense, previous studies have proposed the ecosystem functional biogeography concept, which integrates the effects of plant traits and environmental conditions in assessing ecosystem function[9].

In ecology, the trait-based approaches offer a promising way to generalize predictions across organizational and spatial scales, independent of taxonomy. Accordingly, predicting ecosystem processes and functions such as GPP from functional traits instead of species identity has been considered the "holy grail" of trait-based ecological studies[10,11]. Although the use of plant traits to capture and predict the variation in ecosystem primary productivity (i.e., GPP) along a broad environmental gradient has aroused widespread interest[10,12–15], a recent study has shown that plant traits alone are poor predictors of ecosystem functions[16]. Most related studies have established correlative linkages between the means of plant traits at the leaf scale and ecosystem primary productivity per unit land area[17–20]. While these studies have provided important insight into the trait-productivity relationship, there remain knowledge gaps[18,21]. Indeed, there is little to no evidence of causal linkage between the mean trait values characterizing leaf-level photosynthesis and total carbon absorption via the continuous activity of all photosynthetic tissues per unit land area during the growing season[18,21]. A high photosynthetic rate per unit leaf area or mass provides limited information about the carbon uptake of the entire plant[18,22], and ignores the carbon-capture capacity of the ecosystem[21]. Even at lower values of leaf-level traits (e.g., lower leaf nutrient concentration), a community's primary productivity per unit land area may still increase[23,24]. A more robust and mechanistic approach is therefore needed to integrate plant traits to predict variation in ecosystem primary productivity along broad environmental gradients.

Recently, a trait-based productivity (TBP) theory, which scales plant traits to the community level, has been proposed (Fig. 1; Text S1). The TBP theory assumes that ecosystem primary productivity is determined by environmental factors, trait quantity (**Trait$_{quantity}$**), trait efficiency (**Trait$_{efficiency}$**), and growing season length (**GSL**). GPP is affected by environmental factors (including growing season temperature, precipitation, and the moisture index) that both power ecosystem carbon-uptake as energy inputs (i.e., affecting net photosynthesis or maintaining respiration) and regulate plant carbon distribution, such as the assimilation of C as a nonstructural compound (i.e., in reserve pools) that represents storage at the expense of organ formation[24]. Trait$_{quantity}$, which standardize traits on the unit land area, represents ecosystems carbon uptake capacity[21]. GSL determines the effective period of carbon absorption in the ecosystem, thus positively influencing GPP[25]. In addition, environmental factors affect GPP both

directly and indirectly, by affecting plant community traits[3,13,26]. We therefore assume that a large part of their effect on GPP is mediated by plant community traits.

Integrating multiple functional traits to predict ecosystem primary productivity, rather than simply relying on the type of trait selected, is a major aspect of the TBP theory[27]. Based on the mass ratio hypothesis, community-weighted mean values of leaf traits (efficiency traits such as leaf nutrient concentration, LNC, and specific leaf area, SLA) are considered to be closely related to ecosystem efficiency[13,28], ultimately affecting GPP[13]. However, although leaf size appears to be a good predictor of GPP at large scales[20], leaf nutrient concentration is not a stable predictor[18,19,22], even though both are based on community-weighted means. These knowledge gaps require a more reliable method to integrate multiple traits to predict GPP, with more convincing clear explanations. Meanwhile, "synthetic traits" (i.e., quantity traits) providing more contextual information (i.e., relating to phenotypic, environmental, and biogeographic context) are better predictors of ecosystem function[18,21]. Synthesizing these concepts, we propose a conceptual GPP model that integrates multiple traits based on TBP theory, representing the proposed hypothesis that the environment, plant community traits (quantity and efficiency traits), and GSL jointly determine GPP (Fig. 2a; Text S1).

We systematically applied a high-quality dataset of plant traits and GPP, spanning broad environmental gradients, to verify our conceptual model based on the TBP theory. We surveyed 72 typical natural ecosystems across China, with high biodiversity and GPP gradients, measuring multiple leaf traits that are closely related to photosynthesis[24,29], and using more than 13,000 plant samples and ca. 2,500 species (Fig. 2b). We asked three primary research questions: 1) How well can structural equation modeling based on TBP theory predict the observed yearly and monthly GPP along broad environmental gradients? 2) How do environmental factors and traits directly and indirectly affect GPP variation? 3) What is the relative importance of environmental factors and traits in determining the variation in GPP?

## Results

Overall, our structural equation modelling (SEM), based on TBP theory, significantly captured GPP variation along broad environmental gradients ($R^2 = 0.87$; Fig. 3a). Even after removing the effect of GSL as a phenological trait (GPP$_{yearly}$ divided by GSL to obtain GPP$_{monthly}$), the overall prediction ability was still strong ($R^2 = 0.73$; Fig. 3b). Intriguingly, Trait$_{quantity}$ had the highest direct effect on GPP$_{yearly}$ ($\beta_{std} = 0.33$; Fig. 4a), while MI$_{gs}$ exerted the highest indirect effect on GPP$_{yearly}$ ($\beta_{std} = 0.48$; Fig. 4a). As

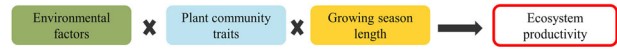

**Fig. 1 An introduction to trait-based productivity (TBP) theory.** As an analogy to the Production Ecology Equation, we use emergent thinking to elucidate the formation of productivity at the ecosystem level, applying several simple and powerful parameters to predict ecosystem productivity (gross primary productivity, GPP). Here, *environmental factors* refers to energy input, representing the total supply of resources in an ecosystem; as *plant community traits*, Trait$_{quantity}$ represents resource uptake and carbon fixation, and Trait$_{efficiency}$ represents the intrinsic efficiency of resource utilization and production; and *growing season length* represents the period of $CO_2$ absorption by the ecosystem. These three key parameters, reflecting ecosystem attributes, jointly determine ecosystem productivity. We extend this analogy multiplicative framework via structural equation modeling, to distinguish direct and indirect effects. In essence, trait-based productivity (TBP) theory scales traits to the community level, then uses plant community traits to predict GPP.

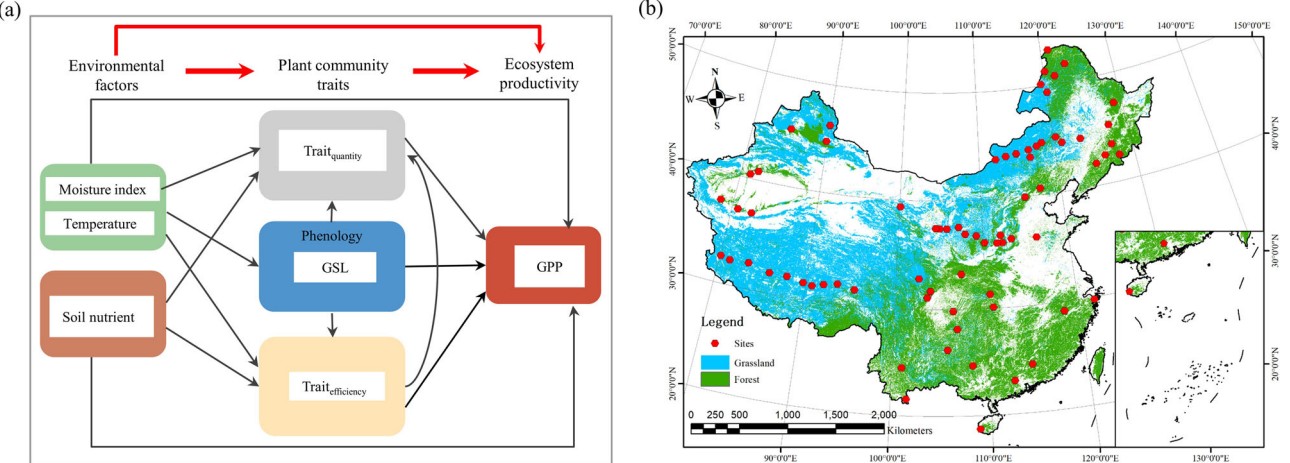

**Fig. 2 Technology roadmap including conceptual model and sampling sites distribution in thie study.** Conceptual model depicting linkages among the environmental factors and plant community traits affecting the variation in GPP (**a**) and geographic distribution of study sites (**b**).Trait$_{quantity}$ and Trait$_{efficiency}$: quantity and efficiency traits, respectively; GSL: growing-season length (months). The map of sample distribution was created using ArcGIS software v10.8.

expected, a large proportion of the impact (71%; Fig. 4b) of the environmental factors [growing season moisture index (MI$_{gs}$), growing season monthly mean temperature (T$_{gs}$), and Soil$_{pc1}$] on GPP$_{yearly}$ was due to indirect effects on GPP$_{yearly}$ via their effects on plant community traits.

Our random forest trait model revealed that the quantity traits predicted the variation in GPP well (Text S2; Fig. S1). The quantity traits (leaf area index, LAI; leaf biomass per unit area, LMI; and total leaf nitrogen and phosphorus per unit land area, LNI and LPI, respectively), were significantly positively associated with GPP$_{yearly}$, and leaf area (LA) and leaf dry mass (LM) were important as efficiency traits (Fig. S1a). For GPP$_{monthly}$, all four quantity traits (LAI, LMI, LNI, and LPI) had robust predictive power, considerably outperforming the efficiency traits (Fig. S1b). Our random forest modeling and forecasting revealed that GPP$_{yearly}$ (Slope, 1.12 ± 0.10, RMSE = 262.41) and GPP$_{monthly}$ (Slope, 1.16 ± 0.16, RMSE = 39.79) were well-predicted, based on TBP theory (Fig. S2).

We applied independent effects analysis (IEA), an important complementary analysis to SEM, to assess the independent effects of each variable on GPP. Based on the GPP$_{yearly}$ model, trait factors accounted for 69% of all the variables' influence on GPP$_{yearly}$, while environmental variables accounted for 31% (Fig. 5). Based on the GPP$_{monthly}$ model, trait factors accounted for 70% of all the variables' influence on GPP, while environmental factors accounted for 30% (Fig. 5). Irrespective of the modeled response variable, the key results did not change when the models were run separately for each vegetation type (Fig. 5).

## Discussion

The primary goal of this study was to test whether this conceptual model based on TBP theory can be used to capture the spatial variation in ecosystem GPP along a wide environmental gradient, at both yearly and monthly scales. Our model successfully captured the spatial variation in both yearly and monthly productivity. Our TBP theory explains clearly how four key controlling elements—environmental factors (T$_{gs}$ and MI$_{gs}$), Trait$_{quantity}$, Trait$_{efficiency}$, and GSL—capture the spatial variation in GPP.

T$_{gs}$ and MI$_{gs}$ regulate plant structural growth by mediating their energy and resource (water and nutrient) inputs[24,30] and carbon allocation[24], thereby collectively affecting GPP. Here,

environmental factors significantly impacted GPP, both directly and indirectly. Previous studies have assumed such a trait-based response-and-effect framework[26], in which plant traits mediate the effects of environmental factors on ecosystem function. Nonetheless, our study is one of few to present empirical evidence of the irreplaceable mediatory role of plant traits.

Our findings for Trait$_{quantity}$, the product of mass-based leaf traits and leaf biomass per unit area (LMI), were consistent with our expectations; this variable substantially affected GPP, even more so than Trait$_{efficiency}$. This is consistent with the carbon economy theory, which predicts that the plant relative growth rate is determined by the biomass allocated to photosynthetic tissues[22,31]. Compared with Trait$_{efficiency}$, Trait$_{quantity}$ represents the vegetation nutrient stocks of the whole ecosystem and thus can be better used to predict GPP. Although the plant nutrient pool is not a direct measure of ecosystem function (i.e. ecosystem fluxes of energy and matter)[32], it is an important attribute that determines ecosystem function (e.g., decomposition, carbon sequestration, nitrification and nutrient recycling)[33–37]. Numerous studies have shown that plant nutrient pool is particularly relevant to the long-term net ecosystem balance of energy and matter[34,37]. Higher plant nutrient pool values mean that, for each unit area of land, the vegetation has better resource utilization capacity, indicating that more productive species are selected (such as leaf area index[38]). Plant nutrient pool is thus considered a driver of ecosystem production-related function[21,39].

Trait$_{efficiency}$, a conventionally used community-weighted trait mean trait, was positively related to GPP; this relationship was primarily driven by LA (see Fig. 5). LA, representing leaf size, is closely related to plant energy balance (including energy uptake and conversion) and is a reliable indicator of GPP on a large scale[20]. A recent global-scale study revealed a strong relationship between LA and canopy size (including canopy height, diameter, and tree height), which reflects the total photosynthetic capacity of the whole tree[40]. While this explains why LA can predict large-scale variation in GPP, this coupling does not necessarily reflect causality. The efficiency traits leaf nutrient concentration (LNC) and leaf phosphorus concentration (LPC), for instance, reflect the photosynthetic rate per unit leaf area or unit leaf mass; however, they do not always link well causally with GPP, because of the absence of other community context information, such as leaf biomass allocation or total leaf area[18,22]. The choice of efficiency traits is therefore critical for predicting ecosystem function[27]. In

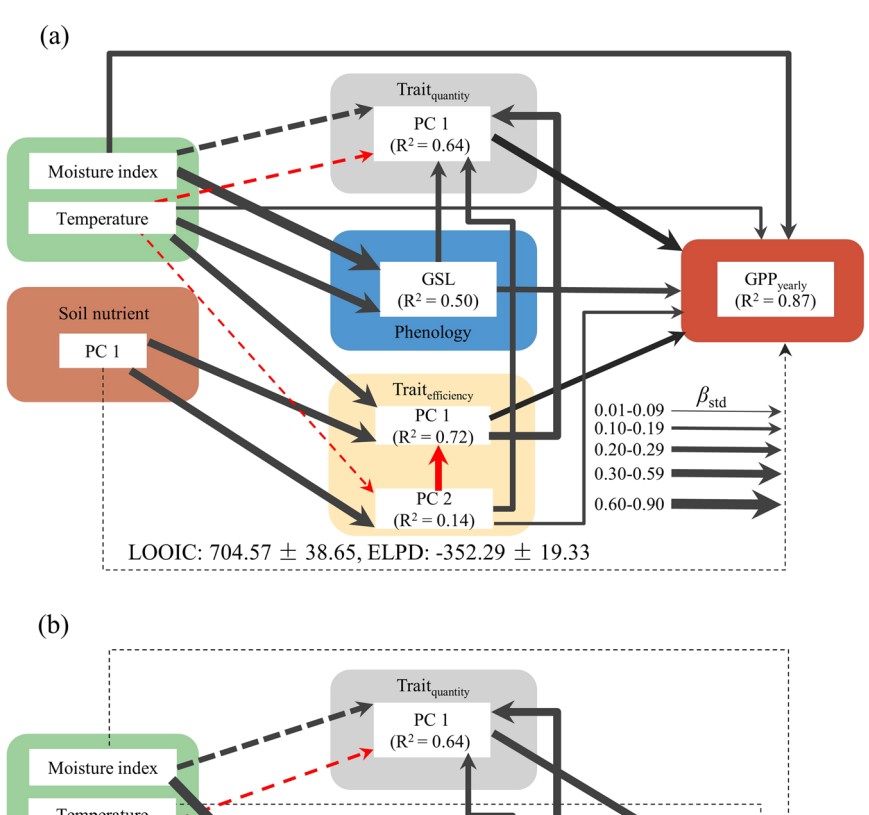

**Fig. 3 Results from Bayesian piecewise structural equation models.** Bayesian piecewise structural equation models exploring the direct and indirect effects of environmental factors and traits on gross primary productivity (GPP) across all sites: (a) yearly GPP (GPP$_{yearly}$) and (b) monthly GPP (GPP$_{monthly}$). PC 1 and PC 2: the first two principal components (PCs) of the corresponding variables; Trait$_{quantity}$ and Trait$_{efficiency}$: quantity and efficiency traits, respectively; GSL: length of the growing season (months); $\beta_{std}$: standardized path coefficients; black and red arrows: positive and negative relationships, respectively; solid and dotted lines: significant and non-significant effects, respectively; arrow width is proportional to the strength of the relationship; LOOIC: leave-one-out cross-validation information criterion; ELPD, expected log predictive density.

contrast, as quantity traits, nutrient concentration and leaf size per unit land area predict variation in GPP well, by standardizing the unit land area[21], making the effect of trait choice less important. GSL plays an important role in shaping GPP by affecting the period of photosynthetic activity[6,25,41].

Notably, Trait$_{quantity}$ and Trait$_{efficiency\ pc2}$ (mainly representing SLA), played key roles at the monthly scale (Fig. 4b), indicating that once the vegetation growth of a particular ecosystem starts under normal resource supply conditions, its efficiency, and especially its capacity to capture resources, becomes more important. SLA, an important component of the plants' relative growth rate model[42], may further affect monthly ecosystem productivity by affecting specific primary productivity[43]. More than 90% of the spatial variation in annual GPP is determined by the CO$_2$ uptake period (i.e., GSL) and the ecosystem's CO$_2$ uptake capacity[25], which are closely related to vegetation community structural variables such as total leaf area and above-ground

biomass[8]. Here, trait quantity, which represents ecosystem CO$_2$ uptake capacity, reflects the structural characteristics of the vegetation community[21]. Therefore, GPP at the monthly scale (GPP$_{yearly}$/GSL) was most affected by Trait$_{quantity}$. Plant growth peak sampling to measure plant functional traits ignores the temporal variation of plant functional traits, which is an important reason that only part of GPP variation can be captured. Seasonal sampling of vegetation to measure plant functional traits combined with dynamic flux observations will help us capture the variation of GPP more accurately. Diversity can enhance total leaf area (i.e., Trait$_{quantity}$) and light interception due to increased canopy packing, thus increasing the likelihood of overyielding[44].

A large component of the impact of environmental factors on ecosystem primary productivity was indirect, via the trait variables (Fig. 5). While there is no doubt that environmental factors significantly affect GPP[4], several studies have proposed quantifying the pathways whereby abiotic factors (environmental

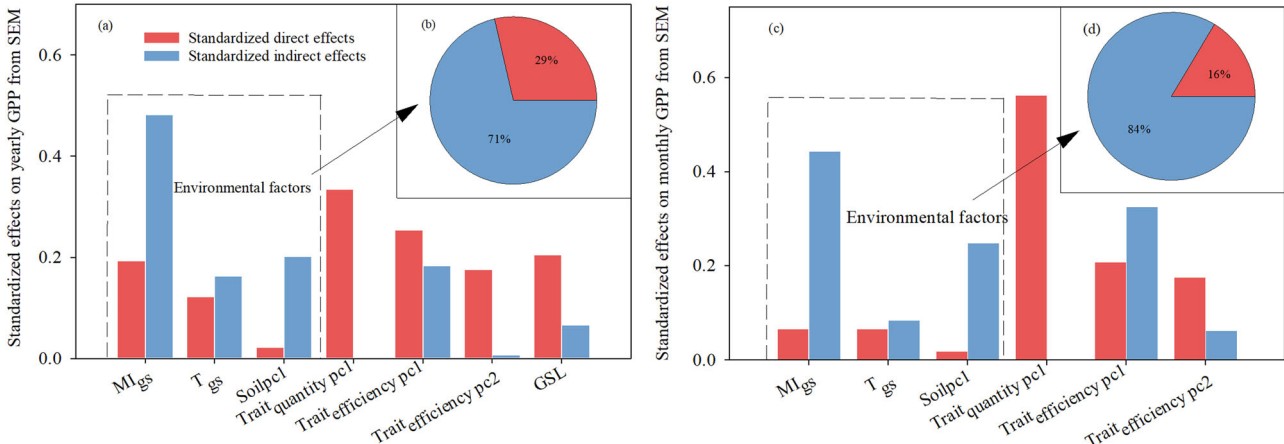

**Fig. 4 Standardized direct and indirect effects of environmental factors and traits on gross primary productivity (GPP) across all sites.** Panels **a** and **b** represent the effect on yearly GPP; **c** and **d** represent the effect on monthly GPP; Panels b and d also additionally represent the ratio of direct and indirect effects of environmental factors on GPP. PC 1 and PC 2: first two principal components (PCs) of the corresponding variables; $Trait_{quantity}$ and $Trait_{efficiency}$: quantity and efficiency traits, respectively; GSL: length of the growing season (months). $MI_{gs}$ and $T_{gs}$: growing season moisture index and temperature, respectively.

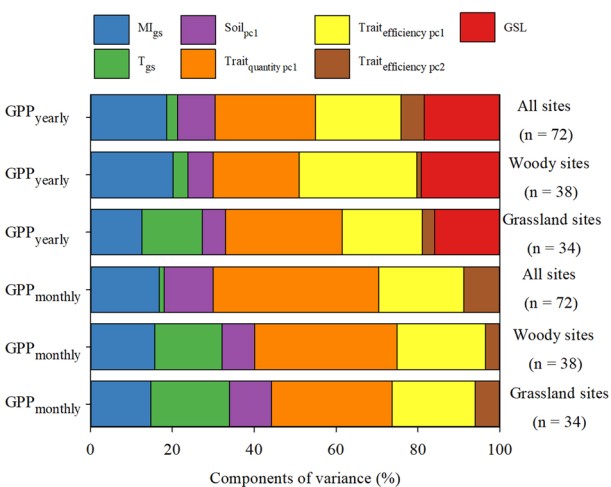

**Fig. 5 Variance partitioning of the independent-effects analysis applied to the annual and monthly gross primary productivity ($GPP_{yearly}$ and $GPP_{monthly}$) models.** $GPP_{yearly}$: yearly GPP model ($GPP_{yearly}$ ~ $MI_{gs}$ + $T_{gs}$ + $Soil_{pc1}$ + $Trait_{quantity}$ + $Trait_{efficiency}$ + GSL). $GPP_{monthly}$: monthly GPP model ($GPP_{monthly}$ ~ $MI_{gs}$ + $T_{gs}$ + $Soil_{pc1}$ + $Trait_{quantity}$ + $Trait_{efficiency}$). PC 1 and PC 2: first two principal components (PCs) of the corresponding factors; $Trait_{quantity}$ and $Trait_{efficiency}$: quantity and efficiency traits, respectively; GSL: growing season length (months). $MI_{gs}$ and $T_{gs}$: growing season moisture index and temperature, respectively. n values in brackets: number of observations for the corresponding group.

factors) contribute to GPP through biotic factors[1,15,41]. Understanding this impact has great significance for quantifying the impact of environmental change on ecosystem carbon cycles[3], especially in the context of biodiversity loss due to global change. Here, particularly at the monthly scale, environmental factors strongly affected GPP, via their effects on traits (Fig. 4b).

These findings have important implications for future research. Much of the impact of environmental factors on GPP is mediated by traits, including $Trait_{quantity}$, $Trait_{efficiency}$, and phenology (growing season length). Plant traits directly affect ecosystem productivity, from first principles. In studies in which traits are not standardized by unit land area[7,17], the indirect effects of environmental factors on GPP are difficult to detect or are greatly

weakened. Indeed, "synthesis traits", such as $Trait_{quantity}$, provide extra information on the total amount of photosynthetic tissue; such traits are needed to improve the capture of multi-dimensional variation in plant function[22]. Here, soil nutrients had no direct effects on GPP, possibly due to scale effects[45], whereas they acted indirectly via their effects on plant traits.

Global changes, such as increased atmospheric $CO_2$ concentration and nitrogen deposition, alter plant biochemical traits (e.g., leaf nutrient concentration)[46] and biomass allocation characteristics[23], which in turn have an important impacts on ecosystem carbon uptake[23]. Our study demonstrates that such plant traits can effectively predict GPP. Therefore, detecting changes in traits will help to elucidate, even predict, potential changes in ecosystem carbon balance, as highlighted in the trait-based response-and-effect framework[26]. The effects of plant functional traits on GPP are not limited to leaf traits. Root traits, such as fine-root nutrients or biomass[38], could potentially influence GPP, especially if they are not coupled or coordinated with leaf traits[47]. Phenological trait datasets require updating. As this study was limited to large-scale data, we could only roughly estimate the length of the growing season. It is better to use first-hand observational data, based on ecological field stations[48]. Further studies are required to identify interactions among the various types of plant community traits and/or across resource gradients. As such, we could better understand or clarify whether $Trait_{quantity}$ and $Trait_{efficiency}$ always act synergistically, and the circumstances under which trade-offs occur. Standardizing traits by unit land area (i.e., $Trait_{quantity}$), and further examining their relationships to ecosystem function, deserves further attention, especially given that most ecosystem functions are quantified on a unit land area basis. This will improve our ability to explain and predict the responses of terrestrial ecosystems to global change at different scales.

## Methods

**Plant traits and soil data**. The dataset of plant functional traits was collected during a large-scale field survey in 72 typical natural ecosystems using a unified sampling standard from 2013–2019. These sites include evergreen broadleaf forests, deciduous broadleaf forests, evergreen coniferous forests, deciduous coniferous forests, shrublands, meadows, steppes, and sparse grasslands, spanning broad environmental gradients and with high environmental heterogeneity. Standardized sampling and measurement protocols were applied to each vegetation and soil surveys. Specifically, the surveyed sites extended from 18.74°N to 53.33°N and 78.47°E to 128.89°E, with mean annual temperatures ranging from −3.8 °C to 22.2 °C, and mean annual precipitation ranging from 32–1942 mm. Plant samples were collected using the quadrat method (30 m × 40 m for the forest, 10 m × 10 m

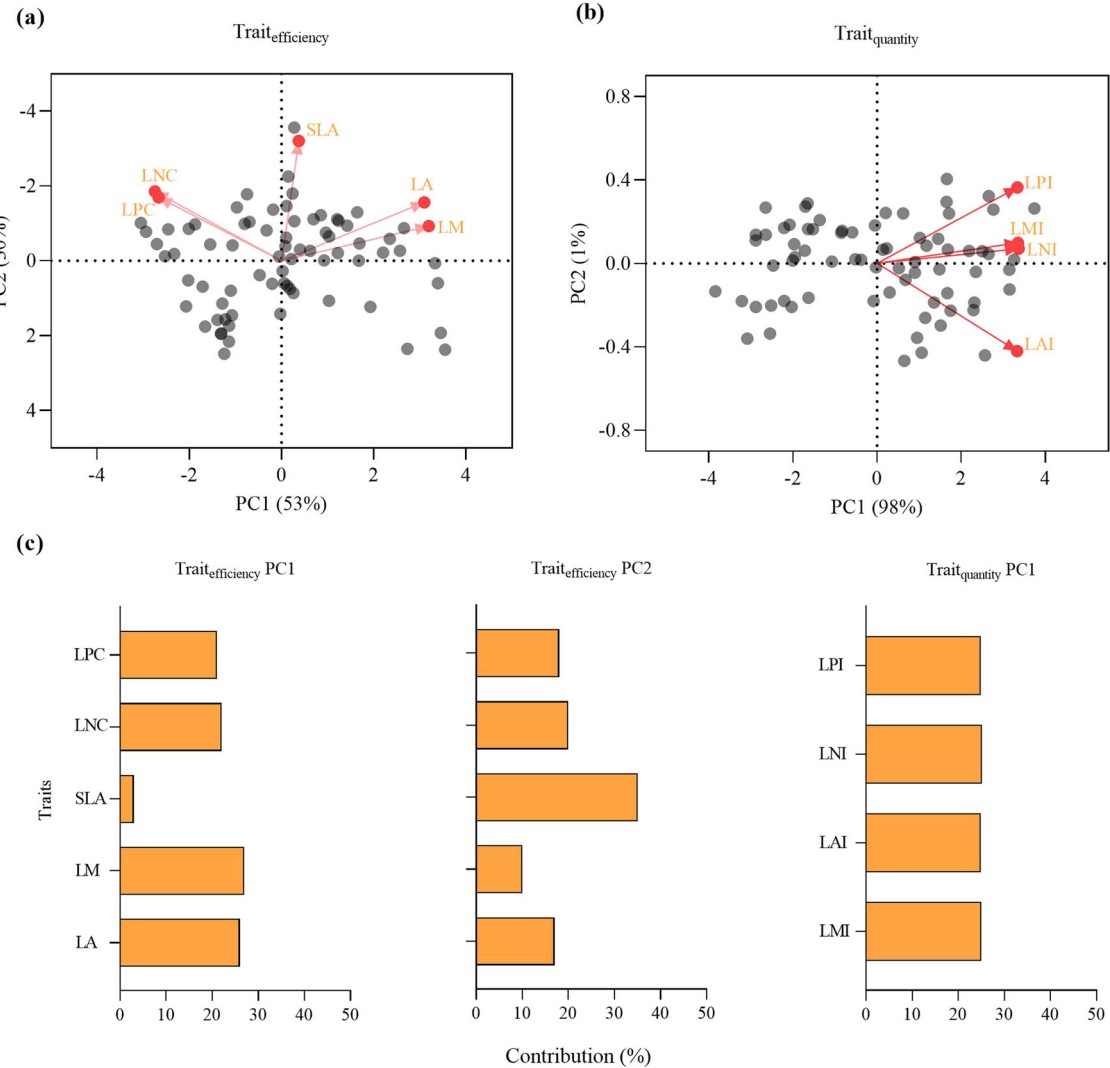

**Fig. 6 Dimensions for multiple traits. a** and **b** Biplots resulting from the principal component analysis for efficiency trait (Trait$_{efficiency}$, **a**) and capacity trait (Trait$_{quantity}$, **b**). Grey points: different sites. **c** Bar plots of the contributions; LA, leaf area (cm$^2$); LM, leaf dry mass (g); SLA, specific leaf area (cm$^2$/g); LNC, leaf nitrogen concentration (mg/g); LPC, leaf phosphorus concentration (mg/g); LAI, leaf area index (m$^2$/m$^2$); LMI, leaf mass index (g/m$^2$); LNI, total leaf nitrogen per unit land area (g/m$^2$); and LPI, total leaf phosphorus per unit land area (g/m$^2$).

for shrubland, and 1 m × 1 m for grassland) to investigate the community structure during the plant growth peak period from July to August (see Text S3 for more information on the sampling protocol). In each plot within a site, key plant community structure variables were measured, including species identity, species number, plant height, diameter at breast height (DBH; basal stem diameter for shrubs) for all woody plants with DBH ≥ 1 cm, and aboveground biomass for herbaceous species. The measured individual-level functional traits for woody and herbaceous plants included leaf area (LA, cm$^2$), leaf dry mass (LM, g), specific leaf area (SLA, cm$^2$/g), leaf nitrogen concentration (LNC, mg/g), and leaf phosphorus concentration (LPC, mg/g), closely related to plant photosynthesis and growth[29,49] (Text S4). Functional traits were divided into size traits, reflecting plant size and light competitiveness, and economic traits, reflecting leaf photosynthetic capacity and nutrient economic[40,50]. All of these traits selected in this study are closely related to the plant light competitiveness and ecosystem photosynthetic capacity. Soil samples from the 0–10 cm soil layer were collected via auger boring for analysis of total soil carbon (%), nitrogen (%), phosphorus (%), and soil pH (Text S5). For further details regarding plot setting, plant trait measurement, and soil analysis, see Text S2–4, and other sources published by this team[51–53].

**Climate and length of the growing season.** Monthly mean temperature (MMT) and precipitation (MMP) data were downloaded from Climatologies at High resolution for the Earth's Land Surface Areas (CHELSA, https://chelsa-climate.org/ )[54,55]. Monthly potential evapotranspiration data were downloaded from the Global Potential Evapo-Transpiration Climate Database (http://www.csi.cgiar.org). The moisture index (MI) was calculated as MMP/PET to represent the monthly water balance of the sample sites[56]. All consecutive months meeting the following

two conditions were determined as months of plant growth: (1) MMT ≥ 5 °C and (2) moisture index (MI) ≥ 0.05[57]. We determined growing season length (GSL) as number of months of plant growth, and further averaged and summed the MI, MMT, and MMP of the growing months (MI$_{gs}$, T$_{gs}$, and P$_{gs}$).

**Plant community traits and gross primary productivity.** As introduced by He, et al.[21,39], we linked GPP by scaling individual-level traits to the community level in two ways:

$$\text{Trait}_{efficiency} = \sum_{i=1}^{n} P_i \times \text{Trait}_i \qquad (1)$$

where $P_i$ is the relative biomass of the $i^{th}$ species in the community (%), and Trait$_i$ represents the leaf nitrogen and phosphorus concentrations of the $i^{th}$ species observed in the plot

$$Trait_{quantity} = \sum_{i=1}^{n} Trait_i \times LMI_i = Trait_{cwm} \times LMI_T \qquad (2)$$

where $n$ is the number of species in the community, Trait$_i$ represents the leaf N and P concentrations of the $i^{th}$ species observed in the plot, $LMI_i$ is the leaf mass per land area of the $i^{th}$ species in a specific community (kg m$^{-2}$), and $LMI_T$ is the total leaf mass per land area in a specific community (kg m$^{-2}$).

Site-specific annual GPP data for 2000–2016 were extracted from a global GPP raster dataset with a moderate spatial resolution (500 m), validated against 113 eddy covariance flux towers across the globe[58]. We also calculated monthly GPP (GPP$_{monthly}$ = GPP$_{yearly}$/GSL).

**Statistics and reproducibility.** To simplify the analysis and avoid overfitting (Fig. S3), a principal component (PC) analysis was performed on all factors (soil factor, trait efficiency, and trait capacity) containing more than three variables (Fig. 6; Table S2). We applied Kaiser's rule to retain PC axes whose eigenvalue is greater than 1, and where the cumulative variance explained by the variables reaches more than 80%, which meets the default variance capture threshold ($\geq$ 70%)[59]. The first two PC axes for the community-weighted means of LA, LM, SLA, LNC, and LPC were used as the Trait$_{efficiency}$ and captured 53% (Trait$_{efficiency\ pc1}$) and 30% (Trait$_{efficiency\ pc2}$) of the variation in these traits, respectively. Trait$_{efficiency\ pc1}$ primarily explained variability in LM (27%), LA (26%), LNC (22%), and LPC (21%). Trait$_{efficiency\ pc2}$ best explained variability in SLA (35%). The first PC axis for leaf area index (LAI, m$^2$/m$^2$), leaf mass index (LMI, g/m$^2$), and total leaf N and P per unit land area (LNI and LPI, g/m$^2$) was used as the Trait$_{quantity}$, and captured 98% of the variation in these traits explaining the variability in the plant community traits evenly: LMI (25%), LAI (25%), LNI (25%), and LPI (25%). The traits were log-transformed before analysis to eliminate size-dependent trait biases[60]. The first PC axis of soil variables (including total soil carbon, nitrogen, phosphorus content, and soil pH) explained 61% of the variation (Table S2). As high collinearity can distort model estimation, collinearity was checked by first calculating the variance inflation factor: this was >5 for P$_{gs}$, which was therefore discarded from the main analysis.

To distinguish the direct and indirect effects of environmental factors and traits on GPP and test our pathway hypothesis (Text S6), Bayesian piecewise structural equation modeling (SEM)[61,62] was used (Fig. 2b). The SEM models fitted in this study were created in the Stan computational framework (http://mc-stan.org/) accessed using the brms package[63] and run with two Markov chain Monte Carlo (MCMC) chains, 10,000 iterations, and a warm-up of 1000 runs. Model convergence was assessed by visually examining trace plots and using $\hat{R}$ values (the ratio of the effective sample size to the overall number of iterations, with values close to one indicating convergence). All $\hat{R}$ values were below 1.01, and effective sample sizes were > 5000 for all coefficient estimates (Supplementary Figs. 2–5). The significance of the coefficient estimates assumes that the credible interval does not include zero. All variables were standardized (mean = 0, standard deviation = 1) before analysis to ensure that standardized path coefficients (hereafter $\beta_{std}$) were obtained. The indirect effect was the product of the direct effects.

LOOIC (leave-one-out cross-validation [LOO] information criterion) and ELPD (expected log predictive density)[64] were used for model verification, using the loo package (for LOOIC and ELPD, smaller and larger values indicate a better fit, respectively)[65]. Posterior prediction checks were performed using the bayesplot package[66]. The validation results for all the models are provided in Supplementary Figs. 2–5. The Pareto shape k is used to diagnose abnormal observation points[64]. Although some of the observations had k estimates reflecting abnormality, the results were not substantially different after removing the outliers (Supplementary Data 1). Considering that the site trait data was obtained via standard protocol field surveys, and the need for caution when deleting outliers, we included all sites and analyses in the main text. The model results without outliers are listed in the attachment (Supplementary Data 1). We repeated the main analysis after dividing the observations into woody and herbaceous communities (Supplementary Data 1), and using a single trait to represent the quantity and efficiency traits, respectively (Supplementary Data 2).

Complementary to the SEM analysis, we used independent effects analysis (IEA) in the R package hier.part to examine the independent contribution of each explanatory variable in predicting GPP[67]. This approach quantifies the contribution of each predictor in explaining total variance in GPP by comparing the fit of all models containing a particular variable to the fit of all nested models lacking that variable, a process referred to as hierarchical partitioning[68]. Considering the correlation between environmental factors and plant traits, and the ability of IEA to robustly partition the independent contributions of correlated predictors, this analysis is highly appropriate and effective for determining the relative importance of the environmental factors and plant traits in our study[68].

**Reporting summary**. Further information on research design is available in the Nature Portfolio Reporting Summary linked to this article.

## Data availability
The data used in this study are available via the Figshare repository (https://doi.org/10.6084/m9.figshare.22081634.v1)[69].

## Code availability
All the R packages used are described in the methods section. No new code (or function) was created.

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

## Acknowledgements

We thank all the members who have participated in field investigation over the eight years to assist us in collecting the data. This study was financially supported by the National Key R&D Program of China (2022YFF080210102), the National Natural Science Foundation of China (42141004, 31988102), by CAS Project for Young Scientists in Basic Research (YSBR-037), and by National Science and Technology Basic Resources Survey Program of China (2019FY101300).

## Author contributions

N.H. and P.Y. designed the research. P.Y., and N.H. conducted the research (collected the datasets and analyzed the data). P.Y. wrote the manuscript. K.V.M., K.Y., L.X., and N.H. commented on the details of the manuscript drafts.

## Competing interests

The authors declare no competing interests.
