## [Peer Review File · Communications Biology]

Reviewers' comments:

Reviewer #1 (Remarks to the Author):

In the abstract, the authors describe some very exciting work that they have carried out, providing accurate quantification of gross primary productivity (GPP) of ecosystems using a trait-based productivity framework. The ability to significantly improve our ability to predict the productivity of ecosystems would be extremely valuable. Unfortunately, I don't find the manuscript to be well enough structured to be confident that the work is of publishable quality. However, I'm hopeful that the authors can clarify this in a subsequent revision.

To clarify my expertise, your reviewer works on ecological and statistical modelling of terrestrial plant ecosystems, and although I have some experience of estimating whole ecosystem productivity, I certainly know there are large gaps in my knowledge.

My fundamental concern is that I'm not clear from the introduction exactly what the trait-based productivity framework is that the authors are using. In the methods it becomes clear that it is based on He et al. (2019) DOI:10.1016/j.tree.2018.11.004, which offers similar results, albeit fitting less well to GPP, and only for forests and not for grasslands. However, this isn't made clear and to add to the confusion, that paper never seems to use the TBP name. I therefore believe that the introduction needs to be significantly improved - at the moment it is just too scattergun in its description, and needs to be better focused to lead to the TBP model that is being used, and how it differs from previous methods.

I also find it hard to understand from the results what has actually been achieved - what the different capacity / efficiency / phenology traits being referred to are. Many acronyms are used before they are explained, and in particular GSL is not explained outside of legends as far as I can see. Again, most of this is comprehensible once you pore through the methods, but this isn't acceptable, particularly given the structure of the paper with the methods at the end. Significant work needs to go into making clear what the results are.

Finally, although I assume the results are better using principal components of the factors involved rather than the raw factors, it would be good to understand what happens with the raw factors as I find it difficult to interpret the meaning of the principal components and the significance of their predicting productivity. I'm also not sure that the paper can talk about a "mechanistic" model, when the inputs to the model are principal components - I'm not sure what the mechanism being referred to is?

I want to stress that this looks like promising work, but I'd really like to see it substantially revised to make it clearer what the TBP framework is, where it comes from, and what the results are.

Reviewer #2 (Remarks to the Author):

The work from Professor He and colleagues nicely presents how a trait-based productivity framework can be used to quantify and predict gross primary productivity (GPP). The authors aimed to leverage plant leaf traits to infer the Spatio-temporal variation of GPP by testing how well the applied framework can predict the yearly and monthly GPP, how environmental factors affect GPP, and how well environmental factors and traits explain GPP across China.

The authors included an impressive dataset of plants and soils spanning six years of field survey taken from a large range of ecosystems. Climate data were obtained from databases. Plant traits used by the authors were Trait_{efficiency} and Trait_{capacity} both related to the nitrogen and phosphorus concentration in leaves (and I guess needles as well). GPP was extracted from a database and calculated at a monthly rate. All data is openly shared to allow researchers to reproduce the work.

The article seems technically sound. My main comment refers to improved readability for researchers outside the field. I provide some major comments below and some line comments.

1. I miss information on how the study could contribute to understanding anthropogenic or climate change impacts on the ecosystem C balance. Furthermore, the research gap on why plant traits are needed to predict ecosystem productivity is not clearly described.
2. The authors could be clearer in their definitions. For example, how is productivity defined? If this is always GPP, please keep the terminology. How are the drivers assessed? I sometimes have the feeling that the definition of productivity varies. For example, in L84, the authors say "plant productivity contributes most to carbon fluxes across ecosystems". Does this refer to GPP? Maybe it is easier to understand if productivity equals GPP and if not, a clear explanation should be provided. This is similar to the definition of "Plant (leaf trait)". It is unknown how it is defined. Is it the photosynthetic activity or the surface area, nutrient content, or nutrient stock? It remains unknown how well the included traits relate to photosynthesis. I think it should be more transparent or directly addressed that Trait efficiency and Trait capacity refer to the leaf nitrogen or phosphorous content. The descriptions should be used continuously throughout the manuscript. Also, the descriptions of environmental factors require clarification. The environmental factors the authors are addressing are not described. And depending on biotic and abiotic changes, the impact on GPP can be drastically different.
4. The discussion often contains very speculative assumptions of processes that were either not tested or that are very general. For example (L247-253), the authors discuss the Trait capacity by generally explaining which information this trait contains and that it is a key parameter for predicting productivity. This is something that should be clear after reading the introduction as it is a prerequisite to use this trait as a predictor for GPP. The following paragraphs are structured similarly. I think the discussion could be improved by focusing more on the research question raised in the beginning and discussing the results obtained from the data evaluation.
4. How are the relations between Trait efficiency and Trait capacity accounted for? Is there any interrelation among all the investigated traits? Is there a variation between the single years?
5. The figures should be provided in a better resolution.
6. Methods. How representative is it to sample the vegetation community only in summer for whole-year predictions? Was only mineral soil included or also the organic layer? I guess that there is a large variation in the thickness of the organic layers across the investigated gradient. Were the roots in the grassland soil removed by hand? How well do morphological and biogeochemical traits predict GPP? The advantage of standardizing traits to the unit land area over using the trait itself could be more emphasized as this is special in this study. Information on how soil phosphorus was measured is missing.
7. There are many abbreviations used in the text. Maybe a list of used abbreviations is helpful or a clear clarification at the beginning.

Line comments:

L104: What are smaller trait values?

L112-114: This sentence is not clear. It is unclear what the authors mean by the C distribution of vegetation. The biomass of the vegetation? The sentence concludes that this affects productivity.

L118-124: This sentence is very complicated and hard to understand.

L241: causal explanation for what?

L245-247: It is well proven that climate and soil affect plant growth. Please emphasize why it is special that you could show that with your data.

Reviewer #3 (Remarks to the Author):

The authors present an approach to predicting ecosystem productivity (GPP) via traits using a structural equation modelling approach that allows for both direct and indirect effects of drivers. The authors find that trait based characteristics play a key role in explaining GPP, including at a monthly scale.

I have to say that I very much enjoyed reading this manuscript. I found it to be well written and well structured - including motivating the issue and question of interest. I do have a few points that I would like to see the authors to address before I think it is suitable for publication.

My main interest is in the use of R^2 as the means to justify all conclusions. Can this choice be justified and do the authors believe this to be a robust metric for asserting "predictive performance" in this instance?

I can personally see 2 issues with relying on R^2 and I would be interested to hear the authors' response. Firstly, why not use an out of sample set for testing prediction? There seems to be enough data to do this and in my view would be more robust. Secondly, are the R^2 values dominated by the broad gradients being investigated. R^2 is a relative measure and so if variation in GPP is dominated by land use, say, and traits do very well at describing land use, then this could lead to a high r^2 value. Whereas for any individual prediction of GPP there is still a high degree of residual variation.

What was the rationale behind using principle component axis scores and what was the objective threshold used for retaining axis. This isn't clear at present and it seems as though using 2 axis or 1 was quite subjective. I would like some more information on how this was objectively done and why.

Minor Comments :

Did SEMS check for any redundant links in the proposed network? I think that is always important to do.

I think the authors should be careful throughout the paper about over-emphasizing causal effect - your inference is still conditional on a model and the assumption it is the correct one.

I think the results section could be simplified and made more readable as I was having to flick around a lot and cross reference a lot.

Line 112: GSL has not been introduced.

Figure 1b seems to be wrong

Better labelling on figures are required, for example - I am not clear in fig 1a whether PC1 under soil nutrient is the same as PC1 under trait efficiency. I have to read quite a bit further on to understand this.

Line 166 SEM not introduced

Might just be my version, but the quality of figures was really poor. Resolution needs improving

Point-to-point responses to the comments of reviewers

(COMMSBIO-22-1958-T: Integrating multiple traits to predict ecosystem productivity)

Response to reviews:

(Original comment and query in *Italic* and key comments are underlined; Response in **Calibri**)

To reviewer #1

Reviewer #1 (Remarks to the Author):

In the abstract, the authors describe some very exciting work that they have carried out, providing accurate quantification of gross primary productivity (GPP) of ecosystems using a trait-based productivity framework. The ability to significantly improve our ability to predict the productivity of ecosystems would be extremely valuable. Unfortunately, I don't find the manuscript to be well enough structured to be confident that the work is of publishable quality. However, I'm hopeful that the authors can clarify this in a subsequent revision.

To clarify my expertise, your reviewer works on ecological and statistical modelling of terrestrial plant ecosystems, and although I have some experience of estimating whole ecosystem productivity, I certainly know there are large gaps in my knowledge.

Response: We greatly appreciate this reviewer's positive comment on our manuscript. In the revised manuscript, we have substantially improved the structure and presentation of this study as the reviewer suggested.

My fundamental concern is that I'm not clear from the introduction exactly what the trait-based productivity framework is that the authors are using. In the methods it becomes clear that it is based on He et al. (2019) DOI:10.1016/j.tree.2018.11.004, which offers similar results, albeit fitting less well to GPP, and only for forests and not for grasslands. However, this isn't made clear and to add to the confusion, that paper never seems to use the TBP name. I therefore believe that the introduction needs to be significantly improved - at the moment it is just too scattergun in its description, and needs to be better focused to lead to the TBP model that is being used, and how it differs from previous methods.

Response: Yes, our work is based on previous research done by the team (He et al. 2019; He et al. 2022a). The core idea of this study is to scale plant traits to the community level for capturing spatial variation in ecosystem productivity (**Figure 1**). The plant community trait based approach – TBP are expected to be applicable to both forests and grasslands, as shown in this study. In our recent study, we further gave the specific formula of derivation, and can include the intraspecific trait variation (He et al. 2022a). In essence, the TBP theory is the basis or foundation of this study. As such, to follow the reviewer's suggestion, here we have added a Box to give an introduction or summary of core idea, formula and approach of TBP. As such, the readers could more clearly understand what TBP means.

Figure 1 Integrating plant traits into the prediction of ecosystem functioning. Plant community traits are drivers of productivity in the trait-based productivity theory. Abiotic factors, such as light, temperature, precipitation, and soil nutrients, and biotic factors such as herbivory would not only directly affect productivity, but also interact with traits and, in addition influence traits and thus take on an additional indirect influence on the gross primary productivity (GPP) and net primary productivity (NPP). The figure from He *et al.* (2022b).

I also find it hard to understand from the results what has actually been achieved - what the different capacity / efficiency / phenology traits being referred to are. Many acronyms are used before they are explained, and in particular GSL is not explained outside of legends as far as I can see. Again, most of this is comprehensible once you pore through the methods, but this isn't acceptable, particularly given the structure of the paper with the methods at the end. Significant work needs to go into making clear what the results are.

Response: We deeply regret about the confusion caused by these terms. Indeed, our manuscript has a large number of abbreviations, as the reviewer points out. As such, we added a Glossary to more clearly define each term as suggested by a reviewer 2. The results section were placed at the end of the paper to meet a standard format requirement for the Communication Biology.

Finally, although I assume the results are better using principal components of the factors involved rather than the raw factors, it would be good to understand what happens with the raw factors as I find it difficult to interpret the meaning of the principal components and the significance of their predicting productivity. I'm also not sure that the paper can talk about a "mechanistic" model, when the inputs to the model are principal components - I'm not sure what the mechanism being referred to is?

Response: We apologize for the difficulty or confusion of principal component analysis (PCA). Our research is testing the trait-based productivity (TBP) theory. However, high collinearity among explanatory variables is an unavoidable problem, and too many variables will also increase the complexity of statistical models. So we had to use PCA to compress the raw data for subsequent analysis, as previous studies did (Jing *et al.* 2015; Craven *et al.* 2018; Chu *et al.* 2019). Alternatively, we could use the primary variable in each principal component. The results would be the same as principal component analysis used here.

We have removed the use of "mechanism", as the reviewers have emphasized: too much emphasis on causality and mechanism can confuse potential readers. Also importantly, we

used the raw data to build models and make prediction with the aid of random forest as important supplements, as also suggested by the third reviewer.

I want to stress that this looks like promising work, but I'd really like to see it substantially revised to make it clearer what the TBP framework is, where it comes from, and what the results are.

Response: We thank the reviewer for the kind words. As clarified above, we added box 1 to summarize the TBP framework. Since both traits and environment directly and indirectly influence ecosystem functions, the “TBP framework” in the original manuscript was further revised to “structural equation modelling based on TBP theory”, such as **line 201-203** in the last paragraph of the Introduction: “1) How well can structural equation modeling based on TBP theory predict the observed yearly and monthly GPP along broad environmental gradients?”. In this sense, the structural equation modelling based on TBP theory allows to examine the direct and indirect effects of traits and environment variables on ecosystem functions – GPP.

To reviewer #2

Reviewer #2 (Remarks to the Author):

The work from Professor He and colleagues nicely presents how a trait-based productivity framework can be used to quantify and predict gross primary productivity (GPP). The authors aimed to leverage plant leaf traits to infer the Spatio-temporal variation of GPP by testing how well the applied framework can predict the yearly and monthly GPP, how environmental factors affect GPP, and how well environmental factors and traits explain GPP across China.

The authors included an impressive dataset of plants and soils spanning six years of field survey taken from a large range of ecosystems. Climate data were obtained from databases. Plant traits used by the authors were Trait efficiency and Trait capacity both related to the nitrogen and phosphorus concentration in leaves (and I guess needles as well). GPP was extracted from a database and calculated at a monthly rate. All data is openly shared to allow researchers to reproduce the work.

Response: We thank the reviewer’s positive comments and also for his/or her time in reviewing our manuscript.

The article seems technically sound. My main comment refers to improved readability for researchers outside the field. I provide some major comments below and some line comments.

1. I miss information on how the study could contribute to understanding anthropogenic or climate change impacts on the ecosystem C balance.

Response: Plant functional traits can be used as an indicator to reflect the effects of climate change on ecosystems (i.e. effect traits), and thus to predict changes in ecosystem functions (such as carbon balance, response traits), as highlighted in the trait-based response-and-effect framework (Suding *et al.* 2008). Therefore, plant functional traits can be used to predict potential changes in ecosystem functions (such as ecosystem C balance) driven by climate change factors. We specially added a paragraph to illustrate it in the discussion section, **see line 340-345**: “Global changes, such as increased atmospheric CO₂ concentration and nitrogen deposition, alter plant biochemical traits (e.g., leaf area and leaf nutrient concentration)⁴⁶ and biomass allocation characteristics²³, which in turn have an important impacts on ecosystem

carbon uptake²³. Our study demonstrates that such plant traits can effectively predict GPP. Therefore, detecting changes in traits will help to elucidate, even predict, potential changes in ecosystem carbon balance, as highlighted in the trait-based response-and-effect framework²⁶.”

Furthermore, the research gap on why plant traits are needed to predict ecosystem productivity is not clearly described.

Response: We reshaped the Introduction, especially adding a paragraph to fill the gap, see **line 145-151**: “In ecology, the trait-based approaches offer a promising way to generalize predictions across organizational and spatial scales, independent of taxonomy. Accordingly, predicting ecosystem processes and functions such as GPP from functional traits instead of species identity is considered the “holy grail” of trait-based ecological studies^{10,11}. Although the use of plant traits to capture and predict the variation in ecosystem primary productivity (i.e., GPP) along a broad environmental gradient has aroused widespread interest^{10,12-15}, a recent study has shown that plant traits alone are poor predictors of ecosystem functions¹⁶.”

2. The authors could be clearer in their definitions. For example, how is productivity defined? If this is always GPP, please keep the terminology. How are the drivers assessed? I sometimes have the feeling that the definition of productivity varies. For example, in L84, the authors say “plant productivity contributes most to carbon fluxes across ecosystems”. Does this refer to GPP? Maybe it is easier to understand if productivity equals GPP and if not, a clear explanation should be provided. This is similar to the definition of “Plant (leaf trait)”. It is unknown how it is defined. Is it the photosynthetic activity or the surface area, nutrient content, or nutrient stock? It remains unknown how well the included traits relate to photosynthesis. I think it should be more transparent or directly addressed that Trait_{efficiency} and Trait_{capacity} refer to the leaf nitrogen or phosphorous content. The descriptions should be used continuously throughout the manuscript. Also, the descriptions of environmental factors require clarification. The environmental factors the authors are addressing are not described. And depending on biotic and abiotic changes, the impact on GPP can be drastically different.

Response: We have added a Glossary and Box to specifically explain the terminology and concepts covered in this paper. For example, for Trait_{quantity} we give the definition in Glossary: “Trait_{quantity}: quantity traits, standardize traits on the unit land area and represent the capacity for resource uptake and carbon fixation. The quantity traits included in this study are total leaf area (m² m⁻²), biomass (g m⁻²), nitrogen (g m⁻²), and phosphorus (g m⁻²) content per unit land area.”

For the five basic traits measured at the individual level used in this study, we give further clarifications in the Methods section to illustrate their ecological significance. See **line 374-380**: “The measured individual-level functional traits for woody and herbaceous plants included leaf area (LA, cm²), leaf dry mass (LM, g), specific leaf area (SLA, cm²/g), leaf nitrogen concentration (LNC, mg/g), and leaf phosphorus concentration (LPC, mg/g), closely related to plant photosynthesis and growth^{29,49}. Functional traits were divided into size traits, reflecting plant size and light competitiveness, and economic traits, reflecting leaf photosynthetic capacity and nutrient economic^{40,50}. All of these traits selected in this study are closely related to the plant light competitiveness and ecosystem photosynthetic capacity.”

We reshaped the relevant sentences mentioned by the reviewer for clarity. For example, **line 138-139**: “.....as the primary producer, plants contribute most to carbon fluxes across ecosystems¹.....”. Similarly, “Plant (leaf trait)” is reshaped as “plants, especially its leaves” for clarity. For the environmental factor, we directly give the variables it specifically refers to in

this study in the Introduction. See **line 166-168**: “GPP is affected by environmental factors (including growing season temperature, precipitation, and the moisture index) that both influence ecosystem carbon-uptake as energy inputs (i.e., affecting net photosynthesis or maintaining respiration).....”. Yes, the relationship between biotic factors and GPP can change significantly according to changes in abiotic factors. Therefore, we try to better clarify the definitions and improve the presentation to avoid confusion, as suggested by the reviewer’s comments.

4. The discussion often contains very speculative assumptions of processes that were either not tested or that are very general. For example (L247-253), the authors discuss the Traitcapacity by generally explaining which information this trait contains and that it is a key parameter for predicting productivity. This is something that should be clear after reading the introduction as it is a prerequisite to use this trait as a predictor for GPP. The following paragraphs are structured similarly. I think the discussion could be improved by focusing more on the research question raised in the beginning and discussing the results obtained from the data evaluation.

Response: We appreciate these helpful comments and reshaped the Discussion section based on the reviewer's comments. Specifically, we have removed statements about speculative assumptions and broad discussions without gist that the reviewer specifically noted. For example, see **line 277-289**: “Our findings for Trait_{quantity}, the product of mass-based leaf traits and leaf biomass per unit area (LMI), were consistent with our expectations; this variable substantially affected GPP, even more so than Trait_{efficiency}. This is consistent with the carbon economy theory, which predicts that the plant relative growth rate is determined by the biomass allocated to photosynthetic tissues ^{22,31}. Compared with Trait_{efficiency}, Trait_{quantity} represents the vegetation nutrient stocks of the whole ecosystem and thus can be better used to predict GPP. Although the plant nutrient pool is not a direct measure of ecosystem function (i.e. ecosystem fluxes of energy and matter) ³², it is an important attribute that determines ecosystem function (e.g., decomposition, carbon sequestration, nitrification and nutrient recycling) ³³⁻³⁷. Numerous studies have shown that plant nutrient pool is particularly relevant to the long-term net ecosystem balance of energy and matter ^{34,37}. Higher plant nutrient pool values mean that, for each unit area of land, the vegetation has better resource utilization capacity, indicating that more productive species are selected (such as leaf area index ³⁸). Plant nutrient pool is thus considered as a primary driver of ecosystem production-related function ^{21,39}.”

”.

4. How are the relations between Trait_{efficiency} and Trait_{capacity} accounted for?

Response: Indeed, our study detected a certain correlation between Trait_{quantity} and Trait_{efficiency} ($r = 0.71$, $P < 0.001$). As such, we further discussed and clarified “Further studies are required to identify interactions among the various types of plant community traits and/or across resource gradients. As such, we could better understand or clarify whether Trait_{quantity} and Trait_{efficiency} always act synergistically, and the circumstances under which trade-offs occur.” (**line 350-353**). This phenomenon may stem from the fact that traits at the individual or species level are not independent, but are either coordinated or trade-offs (Díaz *et al.* 2016). And this relationship between traits at the species level persisted at the community level. But obviously, it is beyond the scope of this study, and we’re eager to explore this issue further in future research.

Is there any interrelation among all the investigated traits?

Yes, there are correlations between the different investigated traits. We selected five representative traits in this study, including three economic traits, namely specific leaf area [SLA, cm²/g], leaf N concentration [LN, mg/g], and leaf P concentration [LP, mg/g], and two size traits, namely leaf area [LA, cm²], and leaf dry mass [LM, g]. Traits are not independent of each other (Díaz *et al.* 2016), especially traits of the same type are often highly correlated (ie covariation). With these correlations among traits, we thus used the primary component analysis to examine their effects on GPP.

Is there a variation between the single years?

There should be some changes in plant traits between different years, but the current large-scale research generally ignores the variation of traits on the time scale (Isbell *et al.* 2015; Musavi *et al.* 2017; Oehri *et al.* 2017; Craven *et al.* 2018; García-Palacios *et al.* 2018; Fernández-Martínez *et al.* 2020), especially considering that the productivity of the ecosystem (i.e. GPP) studied is often the average value over many years.

5. The figures should be provided in a better resolution.

Response: We used high resolution (300 dpi) figures instead of the original low resolution figures.

6. Methods. How representative is it to sample the vegetation community only in summer for whole-year predictions?

Response: We thank the reviewer's question. In fact, this is a common practice in ecological research, especially at large scales (Isbell *et al.* 2015; Musavi *et al.* 2017; Oehri *et al.* 2017; Craven *et al.* 2018; García-Palacios *et al.* 2018; Fernández-Martínez *et al.* 2020). We typically conduct field sampling during peak plant growth, as recommended in standard operating practices (Perez-Harguindeguy *et al.* 2016). In theory, GPP can be viewed as a mathematical function of growing season length (GSL) and GPP_{max} (Xia *et al.* 2015), and studies have shown that GPP_{max} plays a more important role than GSL, even determinative in most ecosystem (Zhang *et al.* 2020). Therefore, sampling at the peak of the growing season can help us to quantify the functional trait values of plants in the ecosystem under maximum growth potential (i.e., GPP_{max}). Such functional trait values are also considered to be closely related to GPP.

Of course, this sampling method will inherently ignore the seasonal variation of plant functional traits. This is a potential research direction for us in the future: seasonal sampling of vegetation combined with dynamic flux observations will help us capture the variation of ecosystem productivity on a time scale at a finer scale. However, we acknowledge that this cannot be done depending on our current research conditions and beyond the scope of this study. Some of these points have been clarified in the revised manuscript. See **line 315-320**: "Plant growth peak sampling to measure plant functional traits ignores the temporal variation of plant functional traits, which is an important reason that only part of GPP variation can be captured. Seasonal sampling of vegetation to measure plant functional traits combined with dynamic flux observations will help us capture the variation of GPP more accurately."

Was only mineral soil included or also the organic layer? I guess that there is a large variation in the thickness of the organic layers across the investigated gradient.

Response: Soil sampling occurs in the organic matter layer after removal of litter. Yes, the thickness of the soil organic matter layer varies in different regions, especially in large-scale studies like ours. But in order to unify the operation specification, we can only unify the depth

between different sites.

Were the roots in the grassland soil removed by hand?

Response: All visible roots and organic debris are removed by hand regardless of forest or grassland.

How well do morphological and biogeochemical traits predict GPP?

Response: This comment is very interesting. We discuss this in depth with the reviewers here. Conventionally, traits are grouped into morphological traits, chemical traits and so on. Morphological traits, such as typical leaf size, are positively correlated with GPP (Li *et al.* 2020), while biogeochemical traits, such as leaf nutrients concentration, are poorly correlated with GPP. But not all morphological-type traits have a good relationship with GPP, for example, specific leaf area has a poor relationship with GPP on a large scale.

At present, there is a better classification method for traits on a large scale, which comes from the in-depth statistical analysis of massive trait data, which classifies traits into economic traits and size traits. A previous landmark study on six traits (including plant height, stem specific density, leaf area, leaf mass per area, leaf nitrogen content per unit mass, and diaspore mass) at the global scale identified two main axes of variation: the first reflects the size spectrum of the whole plant and plant organs (i.e. size traits); the second axis corresponds to the "spectrum of leaf economics," which is a measure of plant balance between leaf persistence and plant growth potential (i.e. economic traits) (Diaz *et al.* 2015). Similarly, a recent study considering 18 functional traits of trees also found that variation in these traits was captured by two independent processes: 1) one reflecting tree size and light competition; 2) the other reflecting leaf photosynthetic capacity and nutrient economy (Maynard *et al.* 2021). Overall, size traits had a significant latitudinal pattern (Joswig *et al.* 2022) and were thus well correlated with GPP, whereas economic traits had no clear spatial pattern (Joswig *et al.* 2022) and were poorly correlated with GPP.

The advantage of standardizing traits to the unit land area over using the trait itself could be more emphasized as this is special in this study.

Response: we appreciate the reviewer suggestion. Indeed, this aspect is the soul of our manuscript. As such, we reshaped the Discussion section and emphasized it based on the reviewer's suggestion, see **line 353-356**: "Standardizing traits by unit land area (i.e., Trait_{quantity}), and further examining their relationships to ecosystem function, deserves further attention, especially given that most ecosystem functions are quantified on a unit land area basis. This will improve our ability to explain and predict the responses of terrestrial ecosystems to global change at different scales."

Information on how soil phosphorus was measured is missing.

Response: We added a description of the method for the determination of phosphorus in soil. See appendix 1: "Total soil phosphorus content were determined using an inductively coupled plasma optical emission spectrometer (Optima 5300 DV; Perkin Elmer)."

7. There are many abbreviations used in the text. Maybe a list of used abbreviations is helpful or a clear clarification at the beginning.

Response: We added a glossary as suggested by the reviewer.

Line comments:

L104: What are smaller trait values?

Response: This refers to lower leaf nutrient concentration levels. We added an explanation to this sentence. See **line 159-160**: “Even at lower values of leaf-level traits (e.g., lower leaf nutrient concentration), a community’s primary productivity per unit land area may still increase^{23,24}.”.

L112-114: This sentence is not clear. It is unclear what the authors mean by the C distribution of vegetation. The biomass of the vegetation? The sentence concludes that this affects productivity.

Response: This refers to the fact that environmental factors can affect the distribution process of carbon among different organs of plants. For example, under the influence of environmental stress, assimilative C is stored as a nonstructural compound (i.e., in reserve pools) at the expense of organogenesis, and structural growth is decoupled from photosynthesis. In this case, environmental factors affect GPP not directly by affecting net photosynthesis or maintenance respiration but by regulating carbon allocation. Now the sentence is reshaped as: “GPP is affected by environmental factors (including growing season temperature, precipitation, and the moisture index) that both power ecosystem carbon-uptake as energy inputs (i.e., affecting net photosynthesis or maintaining respiration) and regulate plant carbon distribution, such as the assimilation of C as a nonstructural compound (i.e., in reserve pools) that represents storage at the expense of organ formation²⁴.” (**line 166-170**).

L118-124: This sentence is very complicated and hard to understand.

Response: We reshaped the sentence to keep it clear and concise. Now it reads: “In addition, environmental factors affect GPP both directly and indirectly, by affecting plant community traits^{3,13,26}. We therefore assume that a large part of their effect on GPP is mediated by plant community traits.” (**line 172-175**).

L241: causal explanation for what?

Response: It is inappropriate to place too much emphasis on mechanism and causality, as reviewer 3 pointed out that our study is a confirmatory study based on hypothesis (“your inference is still conditional on a model and the assumption it is the correct one.”). Therefore, the word “causal” was removed. We reshape this sentence as: “Our TBP theory explains clearly how four key controlling elements—environmental factors (T_{gs} and MI_{gs}), $Trait_{quantity}$, $Trait_{efficiency}$, and GSL—capture the spatial variation in GPP.” (**lines 268-270**).

L245-247: It is well proven that climate and soil affect plant growth. Please emphasize why it is special that you could show that with your data.

Response: Similar to the classic trait-based response-and-effect framework for plants (Suding *et al.* 2008), the environmental factors we emphasized here that environmental factors indirectly affect GPP through plant community traits. We specifically highlight this in Figure 4 in the paper through the pie chart. This is an important reason why we used structural equation modeling to carry out this study. We reshape this sentence as: “ T_{gs} and MI_{gs} regulate plant structural growth by mediating their energy and resource (water and nutrient) inputs^{24,30} and carbon allocation²⁴, thereby collectively affecting GPP. Here, environmental factors significantly impacted GPP, both directly and indirectly. Previous studies have assumed such a trait-based response-and-effect framework²⁶, in which plant traits mediate the effects of environmental factors on ecosystem function. Nonetheless, our study is one of few to present

empirical evidence of the irreplaceable mediatory role of plant traits." (lines 271-276).

To reviewer #3

Reviewer #3 (Remarks to the Author):

The authors present an approach to predicting ecosystem productivity (GPP) via traits using a structural equation modelling approach that allows for both direct and indirect effects of drivers. The authors find that trait based characteristics play a key role in explaining GPP, including at a monthly scale. I have to say that I very much enjoyed reading this manuscript. I found it to be well written and well structured - including motivating the issue and question of interest. I do have a few points that I would like to see the authors to address before I think it is suitable for publication.

Response: We thank this reviewer for the positive feedback and appreciate these helpful comments.

My main interest is in the use of R^2 as the means to justify all conclusions. Can this choice be justified and do the authors believe this to be a robust metric for asserting "predictive performance" in this instance?

Response: The reviewer raises a good question. The use of R^2 alone to indicate the predictive performance of the model is indeed very limited, so we add other parameters (i.e. LOOIC, leave-one-out cross-validation information criterion; ELPD, expected log predictive density) to characterize the goodness of fit of our model. At the same time, we used the raw data to build model and evaluation, and gave more parameters, such as RMSE, RMSDE, and slope etc. (see below).

i can personally see 2 issues with relying on R^2 and I would be interested to hear the authors' response. Firstly, why not use an out of sample set for testing prediction? There seems to be enough data to do this and in my view would be more robust. Secondly, are the R^2 values dominated by the broad gradients being investigated. R^2 is a relative measure and so if variation in GPP is dominated by land use, say, and traits do very well at describing land use, then this could lead to a high r^2 value. Whereas for any individual prediction of GPP there is still a high degree of residual variation.

Response: That's a good suggestion. We took the reviewer's suggestion and used the raw data for modeling and validation by means of random forest techniques (**Figure 2**). We use random forest in machine learning for modeling mainly because: 1) This algorithm can handle multicollinearity among explanatory variables; 2) This algorithm can automatically detect the best fit; 3) It reduces overfitting in decision trees and helps to improve the accuracy; 4) Normalising of data is not required as it uses a rule-based approach. Moreover, given that relying solely on R^2 is limited, so we used other metrics, such as Slope, RMSE, and RMSDE, in our new validation. For Bayesian structural equation models, LOOIC (leave-one-out cross-validation information criterion) and ELPD (expected log predictive density) are also provided to indicate the fit of the model (See Appendix Table S1 for more details.).

Figure 2 Observed gross primary productivity (GPP) versus predicted GPP for yearly (a) and monthly (b). Predicted GPP of 30% ($n = 22$) randomly selected sites based on a random forest model obtained using the other 70% ($n = 50$) of the sites. The red dotted line represents the 1:1 line. RMSE, root mean square error; RMSDE, standardized root mean square error.

What was the rationale behind using principle component axis scores and what was the objective threshold used for retaining axis. This isn't clear at present and it seems as though using 2 axis or 1 was quite subjective. I would like some more information on how this was objectively done and why.

Response: We regret that important information was lost in the manuscript regarding the procedure for selecting the principal component axes. Indeed, the reviewer 1 has a similar concern. Here we have added a few sentences to the method section of the manuscript to describe the criteria we have adopted. See **line 411-413**: “We applied Kaiser’s rule to retain PC axes whose eigenvalue is greater than 1, and where the cumulative variance explained by the variables reaches more than 80%, which meets the default variance capture threshold ($\geq 70\%$) 57.”

Minor Comments :

Did SEMS check for any redundant links in the proposed network? I think that is always important to do.

Response: Yes, we checked. Checking for missing paths is an indispensable step in structural equation model analysis. We fully agree with the reviewer's comments. We first used Shipley's test of d-separation, Fisher's C statistic (Shipley 2009, 2013) in the actual analysis to test our conceptual model, which is why there are interactions between different kinds of traits. Bayesian analysis does not provide frequency-school Fischer tests, so we did not provide a note on missing path testing in the Methods section. In fact, we first used Fisher's C statistic to test our conceptual model in the actual analysis to prevent missing paths from causing the model to fail, and this is also why there are interactions between different kinds of traits.

I think the authors should be careful throughout the paper about over-emphasizing causal effect - your inference is still conditional on a model and the assumption it is the correct one.

Response: We appreciate these helpful comments. All statements about causal effects have been removed. Words relating to "mechanisms" associated with causal expressions were also removed.

I think the results section could be simplified and made more readable as i was having to flick around a lot and cross reference a lot.

Response: We condensed the Results section, removing unnecessary and redundant information. The condensed results section now totals 300 words, with three paragraphs of about 100 words each.

Line 112: GSL has not been introduced.

Response: We are very sorry for the loss of information. We have added the full name considering that it appears here for the first time in the manuscript. See **line 164-166**: “The TBP theory assumes that ecosystem primary productivity is determined by environmental factors, trait quantity (**Trait_{quantity}**), trait efficiency (**Trait_{efficiency}**), and growing season length (**GSL**).”.

Figure 1b seems to be wrong

Response: This reviewer may be wondering about the South China Sea part in **Figure 1b** (i.e. the small panel in the lower right corner). In fact this is a necessary element in the map relating to presenting the regions of China according to the Chinese administration. We are very sorry for the misunderstanding caused by this.

Better labelling on figures are required, for example - I am not clear in fig 1a whether PCI under soil nutrient is the same as PCI under trait efficiency. i have to read quite a bit further on to understand this.

Response: We corrected it. The "PC" label in **Figure 1** has been removed. All labelling has been condensed to maintain clarity.

Line 166 SEM not introduced

Response: We have added the full name considering that it appears here for the first time in the manuscript. See **line 210-211**: “Overall, our structural equation modelling (SEM), based on TBP theory, significantly captured GPP variation along broad environmental gradients ($R^2 = 0.87$; Fig. 2a).”.

Might just be my version, but the quality of figures was really poor. Resolution needs improving

Response: All figures are updated to ensure a resolution of at least 300 dpi.

References:

1.
Chu, C., Lutz, J.A., Král, K., Vrška, T., Yin, X., Myers, J.A. *et al.* (2019). Direct and indirect effects of climate on richness drive the latitudinal diversity gradient in forest trees. *Ecology Letters*, 22, 245-255.
2.
Craven, D., Eisenhauer, N., Pearse, W.D., Hautier, Y., Isbell, F., Roscher, C. *et al.* (2018). Multiple facets of biodiversity drive the diversity–stability relationship. *Nature Ecology & Evolution*, 2, 1579-1587.
3.
Díaz, S., Kattge, J., Cornelissen, J.H., Wright, I.J., Lavorel, S., Dray, S. *et al.* (2016). The global spectrum of plant form and function. *Nature*, 529, 167-171.
4.
Diaz, S., Kattge, J., Cornelissen, J.H.C., Wright, I.J., Lavorel, S., Dray, S. *et al.* (2015). The global spectrum of plant form and function. *Nature*, 529, 167-171.

5. Fernández-Martínez, M., Sardans, J., Musavi, T., Migliavacca, M., Iturrate-García, M., Scholes, R.J. *et al.* (2020). The role of climate, foliar stoichiometry and plant diversity on ecosystem carbon balance. *Global Change Biology*, 26, 7067-7078.
6. García-Palacios, P., Gross, N., Gaitán, J. & Maestre, F.T. (2018). Climate mediates the biodiversity–ecosystem stability relationship globally. *Proceedings of the National Academy of Sciences*, 115, 8400-8405.
7. He, N., Liu, C., Piao, S., Sack, L., Xu, L., Luo, Y. *et al.* (2019). Ecosystem Traits Linking Functional Traits to Macroecology. *Trends in Ecology & Evolution*, 34, 200-210.
8. He, N., Yan, P., Liu, C., Xu, L., Li, M., Van Meerbeek, K. *et al.* (2022a). Predicting ecosystem productivity based on plant community traits. *Trends in Plant Science*.
9. He, N., Yan, P., Liu, C., Xu, L., Li, M., Van Meerbeek, K. *et al.* (2022b). Predicting ecosystem productivity based on plant community traits. *Trends in Plant Science*.
10. Isbell, F., Craven, D., Connolly, J., Loreau, M., Schmid, B., Beierkuhnlein, C. *et al.* (2015). Biodiversity increases the resistance of ecosystem productivity to climate extremes. *Nature*, 526, 574-577.
11. Jing, X., Sanders, N.J., Shi, Y., Chu, H., Classen, A.T., Zhao, K. *et al.* (2015). The links between ecosystem multifunctionality and above-and belowground biodiversity are mediated by climate. *Nature communications*, 6, 1-8.
12. Joswig, J.S., Wirth, C., Schuman, M.C., Kattge, J., Reu, B., Wright, I.J. *et al.* (2022). Climatic and soil factors explain the two-dimensional spectrum of global plant trait variation. *Nature ecology & evolution*, 6, 36-50.
13. Li, Y., Reich, P.B., Schmid, B., Shrestha, N., Feng, X., Lyu, T. *et al.* (2020). Leaf size of woody dicots predicts ecosystem primary productivity. *Ecology Letters*, 23, 1003-1013.
14. Maynard, D.S., Bialic-Murphy, L., Zohner, C.M., Averill, C., van den Hoogen, J., Ma, H. *et al.* (2021). Global trade-offs in tree functional traits. *bioRxiv*.
15. Musavi, T., Migliavacca, M., Reichstein, M., Kattge, J., Wirth, C., Black, T.A. *et al.* (2017). Stand age and species richness dampen interannual variation of ecosystem-level photosynthetic capacity. *Nature ecology & evolution*, 1, 1-7.
16. Oehri, J., Schmid, B., Schaepman-Strub, G. & Niklaus Pascal, A. (2017). Biodiversity promotes primary productivity and growing season lengthening at the landscape scale. *Proceedings of the National Academy of Sciences*, 114, 10160-10165.
17. Perez-Harguindeguy, N., Diaz, S., Garnier, E., Lavorel, S., Poorter, H., Jaureguiberry, P. *et al.* (2016). Corrigendum to: New handbook for standardised measurement of plant functional traits worldwide. *Australian Journal of botany*, 64, 715-716.

18.

Shipley, B. (2009). Confirmatory path analysis in a generalized multilevel context. *Ecology*, 90, 363-368.

19.

Shipley, B. (2013). The AIC model selection method applied to path analytic models compared using a d-separation test. *Ecology*, 94, 560-564.

20.

Suding, K.N., Lavorel, S., Chapin Iii, F., Cornelissen, J.H., Díaz, S., Garnier, E. *et al.* (2008). Scaling environmental change through the community - level: A trait - based response - and - effect framework for plants. *Global Change Biology*, 14, 1125-1140.

21.

Xia, J., Niu, S., Ciais, P., Janssens, I.A., Chen, J., Ammann, C. *et al.* (2015). Joint control of terrestrial gross primary productivity by plant phenology and physiology. *Proceedings of the National Academy of Sciences*, 112, 2788-2793.

22.

Zhang, W., Yu, G., Chen, Z., Zhang, L., Wang, Q., Zhang, Y. *et al.* (2020). Attribute parameter characterized the seasonal variation of gross primary productivity (α GPP): Spatiotemporal variation and influencing factors. *Agricultural and Forest Meteorology*, 280, 107774.

REVIEWERS' COMMENTS:

Reviewer #1 (Remarks to the Author):

The authors have addressed my concerns from my initial review well, and as a result I have only relatively minor comments that still need to be addressed, much of which continues to relate to explanation and (now more often) correct citation of prior work.

The glossary at the start is particularly appreciated. GPP is referred to there as an annual quantity but studied as both annual and monthly, so that should be corrected. Migs should also be better explained, and SEM and IEA should have references, either there or more likely in the main text where they are introduced. LOOIC and ELPD are also missing. In general techniques like SEM and Random Forest are introduced without giving credit to the authors - this kind of work should be cited just like the ecosystem productivity literature. Often the packages that provide functionality are cited (which is good for reproducibility), but not the primary literature by the inventors of the technique. There are also no references for the CHELSA data that I can see (just a link to the website) and also for the stan package used for the Bayesian analysis. In general, the authors need to go through the manuscript carefully identifying the techniques and datasets used, and make sure that the primary literature is cited.

These are relatively minor issues, however, and this work provides exciting new results and should be published.

Reviewer #2 (Remarks to the Author):

The authors thoroughly addressed the reviewers' comments and significantly improved the quality of the manuscript. Considering the reviewers' comments that raised fundamental issues such as readability, methods descriptions, and clarification of abbreviations, the authors have done an excellent job of revising all items in a highly professional and convincing manner.